# Fatigue Property and Small Crack Propagation Mechanism of MIG Welding Joint of 6005A-T6 Aluminum Alloy

**DOI:** 10.3390/ma15134698

**Published:** 2022-07-04

**Authors:** Zeng Peng, Shanglei Yang, Zhentao Wang, Zihao Gao

**Affiliations:** 1School of Materials Engineering, Shanghai University of Engineering Science, Shanghai 201620, China; pengzeng98@163.com (Z.P.); wangzhentao23@163.com (Z.W.); gzh_960303@163.com (Z.G.); 2Shanghai Laser Intelligent Manufacturing and Quality Inspection Professional Technical Service Platform, Shanghai 201620, China

**Keywords:** 6005A-T6 aluminum alloy, MIG, fatigue, crack propagation

## Abstract

In this study, metal inert gas welding (MIG) was applied to 4 mm thick 6005A-T6 aluminum alloy welding. Compared with other parts, the hardness of the weld zone (WZ) was the lowest, about 67 HV. There was the Softening in WZ, which might make WZ the weakest zone. Then, fatigue tests were carried out on MIG welded joints. All the fatigue specimens fractured at the weld toe of the lap joint, and the fracture was characterized by a cleavage fracture. Crack closure induced by oxide was observed during the steady propagation of the fatigue crack. Impurities hindered crack propagation, changed the direction of crack propagation, and appeared in stepped fatigue strip distribution morphology; in the process of the main crack propagation, the initiation and propagation of small cracks were easily restricted and hindered by the main crack, which slowed down the propagation rate and even stopped the propagation directly.

## 1. Introduction

With the increasing demand for lightweight in industries such as automobiles and rail transportation, aluminum alloys have become more and more popular and have been widely used [1,2,3]. Because of its good formability, weldability, and corrosion resistance, 6xxx series aluminum alloy welded joint components are widely used in urban rail, high-speed rail bodies and shells. [4,5]. Welded joint components have a wide range of applications in rails. Metal Inert Gas Shielded Welding (MIG) has become the main welding process for different aluminum alloy welded joints due to its fast-welding effect, low cost, and easy operation [6,7]. Meng et al. [8] found that fatigue cracks in MIG’s 6082 aluminum alloy welded joints will continue to propagate forward when they encounter non-deformable second-phase particles, but under the action of variable second-phase particles, they will stop expanding.

In order to ensure the safety and reliability of long-term operation in trains and subways, aluminum alloy materials are required to have good fatigue performance and reliable welding process performance [9,10]. Y. Zedan et al. [11] used continuous and pulsed lasers to study the fatigue properties of AA6005-T6 aluminum alloy by laser mode. It was found that the pulse wave laser mode produces higher fatigue resistance. The fatigue of crystal materials is a local gradual process involving the development and formation of microstructure features at different length scales [12]. Wang et al. [13] performed surface strengthening treatment on 6005A aluminum alloy. It was found that the reduction in porosity will make the grain refinement, reduce crack initiation, and crack propagation, and obtain better fatigue performance. In the field of aluminum alloy fatigue research, the fatigue life of aluminum components mainly depends on crack propagation. Most studies focused on fatigue crack initiation, the crack propagation mechanism and fatigue life prediction, while the related studies on small crack propagation were scarce [14,15,16].

Small cracks generally interact with the microstructure, affecting the growth rate of small cracks [17,18]. There are few studies on how microstructure affects the propagation of small cracks and how other large-scale cracks smile. The focus of this paper is the influence of the main crack propagation on the secondary cracks and the micro mechanism of the weld. The fatigue properties are characterized by optical microscope and scanning electron microscopy. The emphasis is to study the propagation mechanism and influence of the main crack and small crack in the stable propagation zone of the fatigue crack.

## 2. Experiments

### 2.1. Materials

The BM, 6005A aluminum alloy in the T6 condition, was processed into 4 mm thick sheets. The welding wire is ER5356 alloy with a diameter of 1.4 mm. The chemical composition of 6005 aluminum alloy and welding wire is shown in Table 1 [19].

### 2.2. Welding Procedure

Double-sided lap MIG welding was used for 6005 aluminum alloy, and the overlap of the welded joint was 20 mm. In the welding process, the shielding gas is pure argon. The welding parameters are shown in Table 2. SiC paper and acetone reagent were used to remove the alumina coating and surface oil pollution near the welding position before welding.

### 2.3. Microscopic Observations

The metallographic samples were firstly polished with SiC paper with a roughness of 400# to 1200# from high to low. Then the surface of the sample was polished with a polishing machine until it became in a mirror image state. Finally, the metallographic surface was etched by Keller reagent (HF:HCl:HNO_3_ = 2:3:5), and the corrosion time was 60 s. The microstructure of the welded joint was observed under the VHX-6000 ultra-depth-of-field microscope.

### 2.4. Hardness Test

It was tested by an HV-1000 microhardness tester (Shanghai Taiming Optical Instrument Co., Ltd. Shanghai, China) that included the hardness of the weld zone (WZ), heat affected zone (HAZ) and base metal (BM). The load of the hardness test is 0.98 N, and the holding time is 15 s. The hardness was tested every 0.25 mm. Figure 1 shows the hardness test method.

### 2.5. Fatigue Test

According to the national standard GB/T3075-2008 design, the fatigue sample size is shown in Figure 2. The preparation of the fatigue specimen was completed by wire cutting. The fatigue test was carried out by using Zwick/Roell Amsler HB250 electro-hydraulic servo testing machine (Zwick/Roell, Ulm, Germany). The fatigue cycle mode of the tensile load is a sine wave. The loading frequency is 20 Hz, and the stress ratio R = 0.1. The ambient temperature is 299 K and the relative humidity is 55%. The statistical method of fatigue S–N curve complies with ISO 12107:2003 standard. Before the fatigue test, the surface and sides of the sample should be polished with sandpaper. The fatigue fracture was observed with a scanning electron microscope (SEM, Hitachi S-3400N, Hitachi, Tokyo, Japan).

## 3. Results and Discussion

### 3.1. Microstructure Analysis

Figure 3a depicts the microstructure of the entire welded joint. In the picture, there were three different zones of BM, HAZ and WZ (Figure 3b, Figure 3c and Figure 3d are, respectively, partial enlarged views of each area of Figure 3a. Figure 3b was near the center of the weld, where snowflake-shaped dendrites, equiaxed crystals and dendrites appeared. It was due to the large degree of undercooling near the weld zone, which would reduce the restriction on crystal nucleation. Crystal nucleation was more free, and the growth rate of its protruding parts was accelerated, which made it extend into the liquid phase faster. Therefore, more dendrites continued to extend into the liquid phase while it continued to extend twice at the solid–liquid interface. Eventually, snowflake-shaped dendrites were formed. Those are the adjacent areas of the weld and the base metal in Figure 3c,d. In Figure 3c, the crystal grains in the HAZ zone became slender and gradually grew closer to the BM zone. Then, the associated crystals gradually formed. However, the HAZ zone of Figure 3d did not produce the same morphological features as in Figure 3c. This was due to the different crystal grain orientations of the base material itself, which resulted in different epitaxial crystal morphologies.

### 3.2. Microhardness Analysis

Figure 4 clearly describes the three large areas of the weld (WZ), welding heat affected zone (HAZ) and base metal (BM). Line 1 and line 2 in Figure 4a have similar graphical curves, and the hardness distribution shows an upward trend from left to right. In Figure 4b, the line 3 curve can be approximated by a V-shaped curve. The base material (BM) had the highest hardness, with an average hardness value of 83 HV. As the distance from the center of the weld increased, the hardness of the joint gradually increased. This was due to the difference in the microstructure of different positions resulting in different hardness. The 6005A-T6 aluminum alloy was a precipitation-strengthened aluminum alloy, and its strengthening effect was mainly related to the movement between the precipitated phase and the dislocation, which the stress generated by the precipitated phase will hinder the movement of the dislocation. Therefore, the base material had more precipitated phases and higher hardness. The average hardness value in the HAZ zone was 68 HV. This was because the precipitation phase dissolves disappeared, and the hardness value decreased. The average hardness value of the WZ zone was 67 HV. The reason for this phenomenon might be that the precipitation phase coarsening occurred due to the instability of the strengthening phase, resulting in a significant decrease in the hardness value in the region and a softening zone [20].

### 3.3. Fatigue Properties Analysis

Table 3 shows the fatigue data of the 6005A-T6 aluminum alloy welded joint and the base metal. Figure 5 is the fatigue life S–N curve diagram drawn based on the fatigue results of the 6005A-T6 welded joint and the base metal in Table 3. According to Figure 5, the fatigue strength of the base metal was significantly better than the fatigue strength of the welded joint. The fatigue limit of a material can be estimated by calculations. The relationship between the stress amplitude and the number of fatigue cycles is usually expressed in exponential form:(1)$$C=N\cdot S_{a}^{k}$$
where *S_a_* is the stress range, C is the material constant, *k* is the fatigue strength exponent, and N is the number of cycles.

For 6005A-T6 base metal, when the stress reached 140 MPa, the plate sample reached 10^7^ cycles without fatigue cracks. When the stress reached 155 MPa, two base metal samples broke during 10^6^ cycles. As the stress value continued to increase, fatigue cracks appeared in all base metal samples, and the number of cycles was between 10^5^ and 10^6^. Through the linear fitting of the fatigue stress and the number of cycles, the fatigue limit of the base metal was 128.9 MPa. When the stress reached 25 MPa, 6005A-T6 welded joints reached 10^7^ cycles without fatigue cracks. Through linear fitting, the fatigue limit of the joint was 22.9 MPa. Therefore, the fatigue strength of the base metal is significantly better than the fatigue strength of the welded joint.

### 3.4. Fatigue Fracture Analysis

Figure 6 and Figure 7 are the fatigue fracture surface diagrams of MIG welded joint specimens, which are the fracture morphology of specimen number 2 (45 MPa, cycle number 586,235) and specimen number 6 (30 MPa, cycle number 4,076,338). Fatigue crack initiation is mostly caused by defects, which generally occur in areas where the stress is concentrated. The two fatigue sources of the specimen are shown in Figure 6a and Figure 7a. The two fatigue sources were, respectively, distributed in the middle and upper areas of the specimen, and both were generated on the surface of the specimen and gradually expand inward. The herringbone pattern could be clearly seen in the picture. Figure 6c is the cleavage crack growth diagram of the crack growth zone. In Figure 6c, the cleavage steps and the river-like morphology can be clearly distinguished. The crack propagation extended from left to right, but after extending to the dotted line, the pattern of the river changed significantly, and the tributaries of the river were significantly reduced. It was because the two parts of the imaginary line are steps of different heights, and the position of the dotted line might be the position of the large grain boundary. The structure of the grain boundary here was more complicated, and the difference between the grains is very large. The right grain had a large cleavage step, which makes it difficult for the crack to continue to grow forward. The feature of dimple morphology appears in the transient region (when the stress intensity range Δ*k* is large enough). Therefore, it can be judged that the fracture mode is cleavage fracture.

Figure 6b and Figure 7b are both about fatigue strip morphology in the fatigue crack growth region. In Figure 6b, the dotted line was divided into three regions. The middle part bulged upward, and the three areas were obviously not on the same plane. Because those were also different than the fatigue band orientation and morphology of the three regions. The grains in this area are complex and diverse, causing the fatigue bands to be distributed on different planes, and the different grain orientations lead to changes in the normal direction of the fatigue bands and changes in the morphology of the fatigue bands. Figure 7b is a microscopic map of a locally amplified out-of-plane fatigue strip. To better describe the microscopic characteristics of the region II in Figure 7b, a simple model diagram is proposed as shown in Figure 7c. The figure shows the common plastic fatigue bands in the fatigue fracture. The spacing between the fatigue bands was also different, increasing from top to bottom, and generally increased with the increase in the stress intensity factor K. The propagation direction of the secondary crack was perpendicular to the direction of the fatigue band. Because of the complexity of material grains, fatigue bands propagated in different planes with different heights and directions.

Figure 7d is a partial enlarged view of Figure 7b, that there are tiny cracks with a size of about 4 μm beside the large cracks. As shown in Figure 7d (III and IV), the small cracks with a size of about 5 μm were all secondary cracks. The secondary crack could not continue to grow when it extends to a certain size. This might be since the propagation of large cracks restricts the propagation of small cracks and weakens their propagation driving force. Then, it was difficult to continue to expand at a small size, and finally stop the expansion, forming a small crack as shown in the figure [20].

Figure 8 shows the fracture morphology of sample number 2 (45 MPa, 586,235 cycles). Figure 8b,c is the local amplification figures in Figure 8a. There were neatly arranged steps in the figure, and there are holes and cracks below the steps. In Figure 8c, there is a cleavage feature called Wallner Liness. The white arrow points to Wallner Liness, and the black arrow points to the direction of crack propagation. The clear steps can be seen in the figure. This is due to the trajectory where the elastic wave emitted when the crack tip meets a certain defect intersects the crack front in the rapid crack propagation.

The stress level had a great influence on the fatigue damage mechanism of aluminum alloy. In the high-stress region with short cycles, the fatigue damage was mainly caused by the dislocation accumulation and stress concentration at grain boundaries [21]. From the above, it could be seen that the fracture position of the sample in this experiment mainly occurs at the weld toe of stress concentration. The characteristic morphology of fishbone-shaped steps that are regularly arranged in Figure 8b might be caused by mechanical tearing under repeated stress under load. Figure 8d is an evolutionary model diagram. At the newly formed step position, the atoms in the high strain state preferentially dissolved, extending longitudinally along the dislocation line, and gradually forming a small hole. Cavities grew up and crossed each other, leading to crack propagation until fracture.

### 3.5. Mechanism of Fatigue Crack Propagation

Figure 9d is a schematic diagram of a larger long crack. The whole crack starts to sprout from the crack source in the middle and then expands to both sides. The morphology on both sides of the crack was obviously different. One side was a river pattern with different flow directions, and the other side was a step-like morphology of different heights. Figure 9b,e are partial enlarged views of the long crack in Figure 9d. In Figure 9e, the microscopic morphologies on both sides of the crack were quite different. One side was a very flat facet and step morphology, and the other side was a regularly arranged fatigue strip. The degree of crack closure had also changed before and after the position of point A. The crack closure on the left of point A was lower, so the width of the crack was larger; similar features also appeared in Figure 9b. In addition, there were local small steps around the cracks. In this regard, it could be judged to be that oxide and other factors induced crack closure. It would cause repeated contact and wear of the fracture surface during the tensile fatigue process that the moisture-containing environment or the micro-roughness of the fracture surface and resulting in oxide coating the crack tip [22,23].

Many secondary microcracks were found near the larger main crack. Figure 9a is a small crack on the lower side of the main crack. There were obvious secondary cracks during the crack propagation. Figure 9a (V) is a crack image processed. Secondary cracks were generally generated at the crack tip and steadily expanded in two directions. In order to better explain the propagation of secondary cracks, a simple propagation diagram of secondary cracks was proposed as shown in Figure 10. The occurrence of secondary cracks was likely to be caused by the complex and diverse grains in this area and the different grain orientations. The cracks extended to areas with complex grain orientations, hindering the original way of expansion and making it easier to expand. Propagation and secondary cracks were generated from this. The fatigue strip in Figure 9f has an obvious discontinuity. This was because the crack growth was hindered by impurities and the fatigue strip was interrupted here, and the secondary cracks accompanying the fatigue strip also occurred. Corresponding changes. However, impurities or second-phase particles would also affect the secondary cracks. The secondary cracks coexisting with fatigue bands appearing next to the main cracks were not in a uniform direction of propagation, which was obviously different from the secondary cracks in Figure 7d. In Figure 9f (IX), there is an enlarged secondary crack morphology, whose propagation path is not uniform, relatively disordered, has impurities and so on.

In Figure 9c, the grain structure near the main crack had also changed, resulting in some small cracks. As shown in Figure 9c (VII), small cracks appeared near the fatigue strip, and their expansion was Z-shaped. Under the combined action of shear and tensile stress, they expanded along the maximum shear stress angle of 45°; the crack presents V-shaped propagation due to the complex grain size in this region, as shown in Figure 9c (VI). The change of crack propagation direction. The change in the direction of crack propagation meant that more energy would be consumed by changing the path, which indirectly would also improve the performance of the material to a certain extent.

### 3.6. Discussion

When the molten pool is cooled rapidly in the welding process, the vacancy concentration gradient in the grains will make the atoms migrate to the grain boundary rapidly. Nonequilibrium segregation at grain boundaries leads to a significant decrease in solute precipitation [24]. The precipitation phase in the weld zone was greatly reduced, which plays a softening role in the WZ zone, and made the WZ zone show a low hardness value. This led to low mechanical properties of the joint, and easy fracturing at the weld. Post-treatment of welded joints might refine grain size. According to the Hall–Petch relation, grain refinement can improve yield strength [25].

Unbalanced segregation of grain boundaries is beneficial to the preferential attachment of micro voids to grain boundaries. The existence of a brittle intergranular phase can also weaken the grain boundary [26]. When the fatigue crack propagated, it would preferentially propagate to the weak grain boundary, resulting in the above different types of stepped morphology characteristics. In addition, the grains in the weld zone were complex and the orientation was chaotic, and many secondary micro cracks appeared around the fatigue crack propagation. The propagation direction of these tiny cracks is usually inconsistent and disordered.

As shown in Figure 11, it was a secondary crack evolution model. As mentioned in the summary of 3.4 above, when the main crack expanded to a certain stage, it would show an angle of 90° with the direction of principal stress, and some stress would promote the generation of secondary microcracks. Brittle intergranular phases such as Al7Cu2Fe and Al2CuMg weaken grain boundaries by a plastic incompatibility mechanism [27]. When subjected to partial shear stress, the secondary crack initiated at the weak grain boundary, and then expanded between grains. The main crack will hinder the propagation of the secondary crack, and the secondary crack usually only extends a small distance. The stress release caused by other crack propagation led to the decrease in the driving force of the secondary crack propagation, and the continuous propagation of the main crack would reduce or stagnation the propagation rate of the secondary crack [26]. Finally, the secondary crack propagation would stop to a certain extent.

## 4. Conclusions

Based on the research results of the fatigue performance of 6005A-T6 aluminum alloy welded joints, a summary is proposed:Due to factors such as heat input, the precipitation phase in the WZ region was reduced, and the hardness of the WZ region was greatly reduced. Fatigue fracture occurred at the lap joint. Using post-treatment means to optimize the welding samples will give the welded joints better mechanical properties.In the stable crack propagation stage, grain orientation and other factors would hinder the crack propagation and change the crack propagation path. In the stable propagation stage, crack closure induced by oxides was also observed.Three evolution models were proposed: fishbone step evolution (model 1), secondary crack initiation evolution (model 2) and secondary crack evolution in main crack propagation (model 3). Model 1 explained the formation of step shape in crack propagation. Models 2 and 3 explained the propagation and interaction of secondary cracks. Model 3 explained the inhibitory effect of the main crack on the secondary crack. The small crack near the crack was easy to be hindered by the main crack propagation, and it was difficult to continue to when the size was small, the expansion stopped.

## Figures and Tables

**Figure 1 materials-15-04698-f001:**
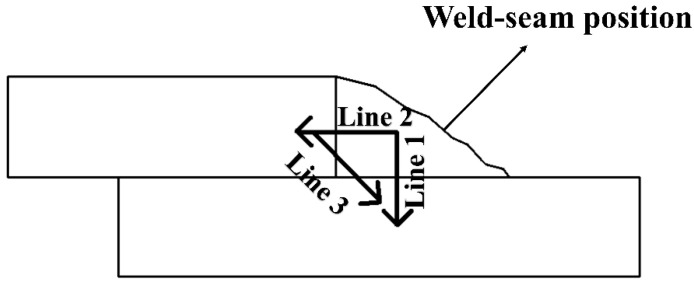
Hardness path diagram.

**Figure 2 materials-15-04698-f002:**
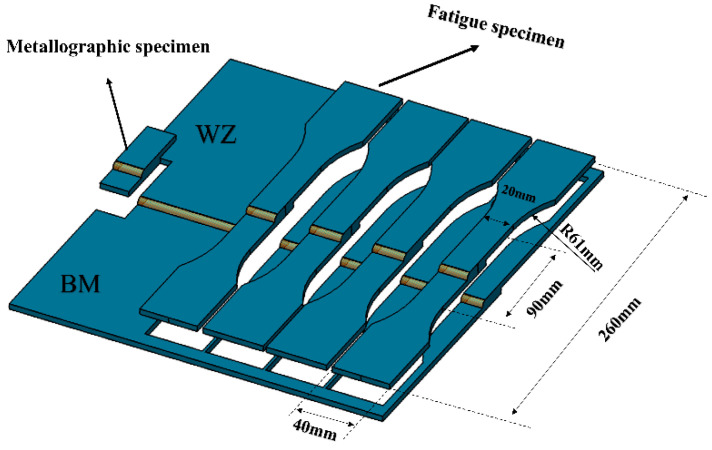
Fatigue test specimens, schematic of the weld joint, and metallographic specimen.

**Figure 3 materials-15-04698-f003:**
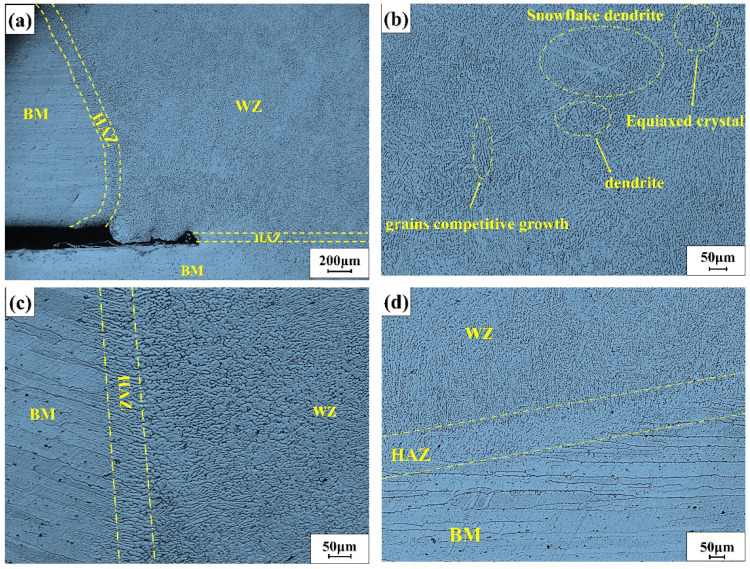
(**a**) The overall morphology of the welded part; (**b**) the microstructure of the weld zone; (**c**,**d**) are the microstructures of the joints in different directions.

**Figure 4 materials-15-04698-f004:**
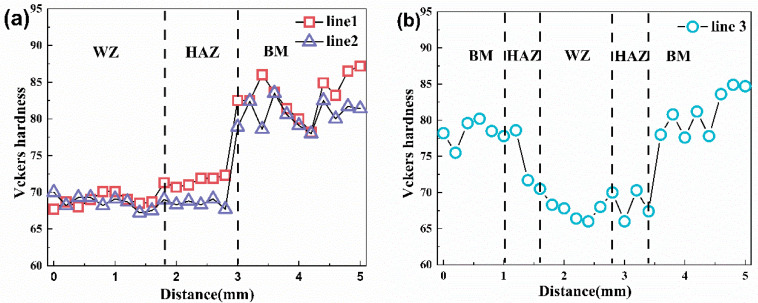
(**a**) The hardness value distribution of line 1 and line 2 in Figure 1; (**b**) Hardness value distribution diagram of line 3 in Figure 1.

**Figure 5 materials-15-04698-f005:**
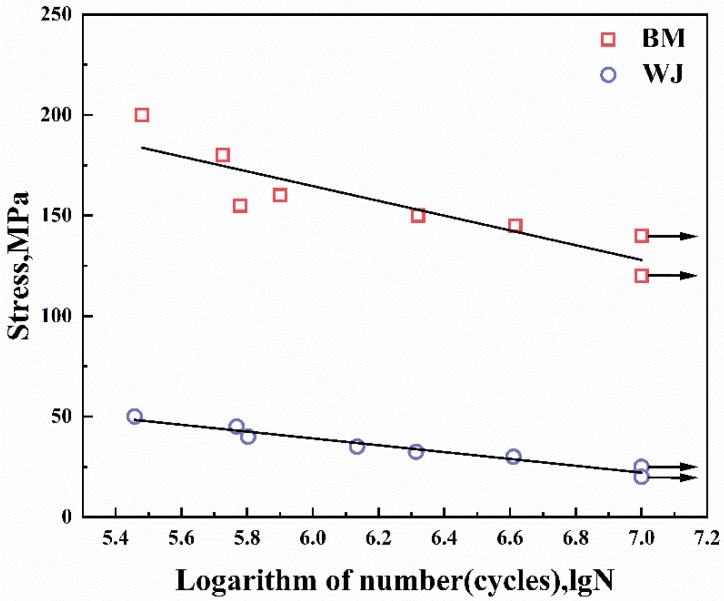
Relationship between stress and fatigue life of welded joint and base metal.

**Figure 6 materials-15-04698-f006:**
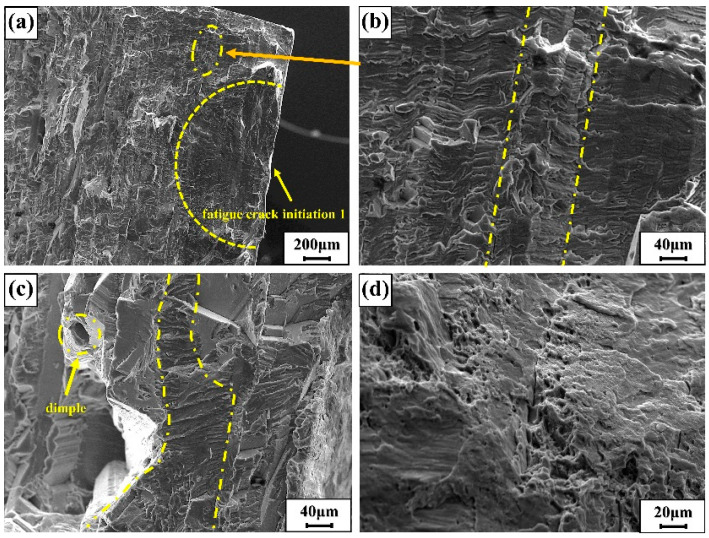
(**a**) Fatigue source zone 1; (**b**) partial enlarged view; (**c**) fatigue crack stable growth zone; (**d**) instantaneous fracture zone.

**Figure 7 materials-15-04698-f007:**
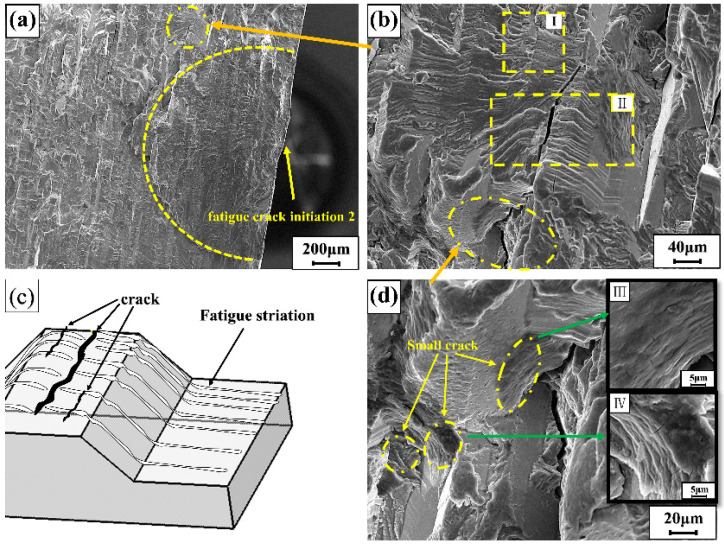
(**a**) Fatigue source zone 2; (**b**) partial high-magnification SEM image; (**c**) model images of cracks and fatigue bands; (**d**) SEM images of cracks at higher magnification.

**Figure 8 materials-15-04698-f008:**
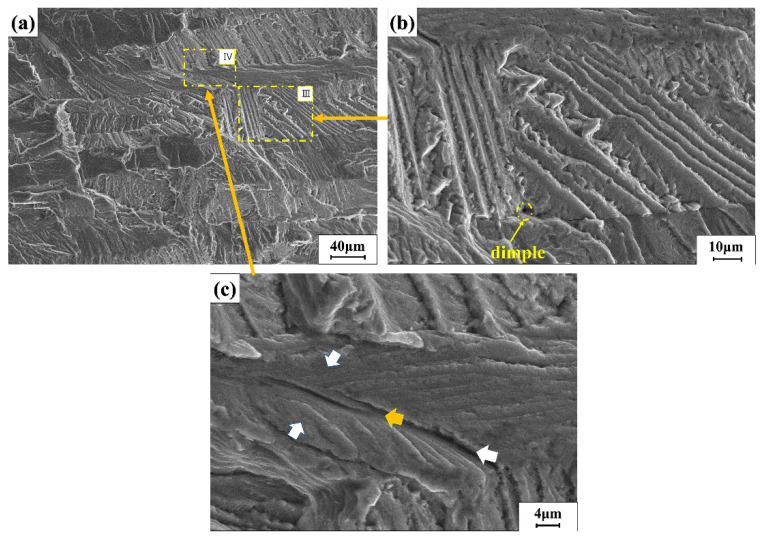
(**a**) The microscopic morphology of the crack propagation zone; (**b**) the partial enlarged view of (**a**) III; (**c**) the partial enlarged view of (**a**) IV; (**d**) evolution model 1.

**Figure 9 materials-15-04698-f009:**
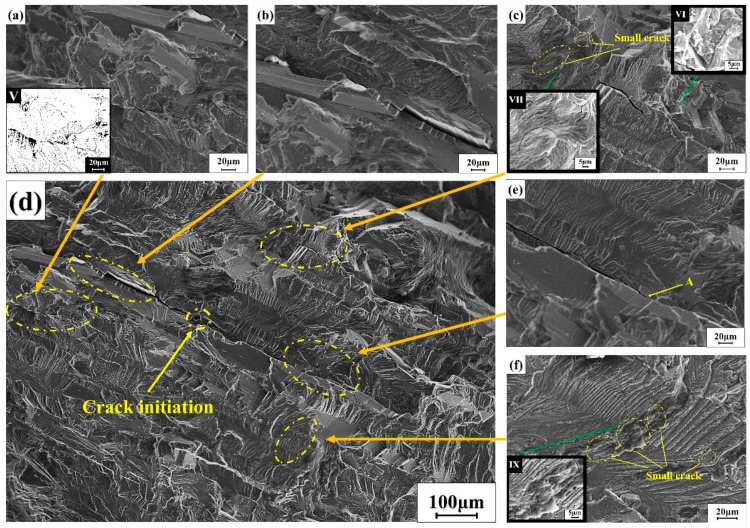
(**a**) Enlarged view 1 of small cracks near the main crack; (**b**) Enlarged view 1 of the main crack; (**c**) Enlarged view 2 of the small cracks near the main crack; (**d**) Microscopic morphology of the main crack; (**e**) Partial enlarged view of the main crack; (**f**) enlarged view 2 of the fatigue band near the main crack.; V is a special processing (**a**); VI, VII, IX were the enlarged area of the green arrow.

**Figure 10 materials-15-04698-f010:**
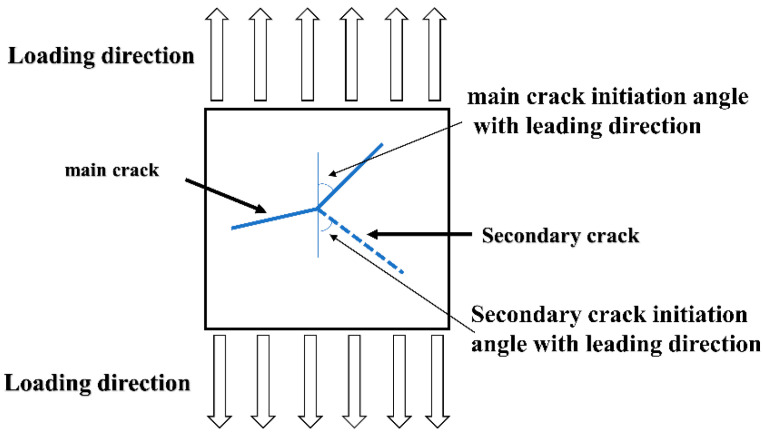
The simplified model 2 of crack propagation.

**Figure 11 materials-15-04698-f011:**
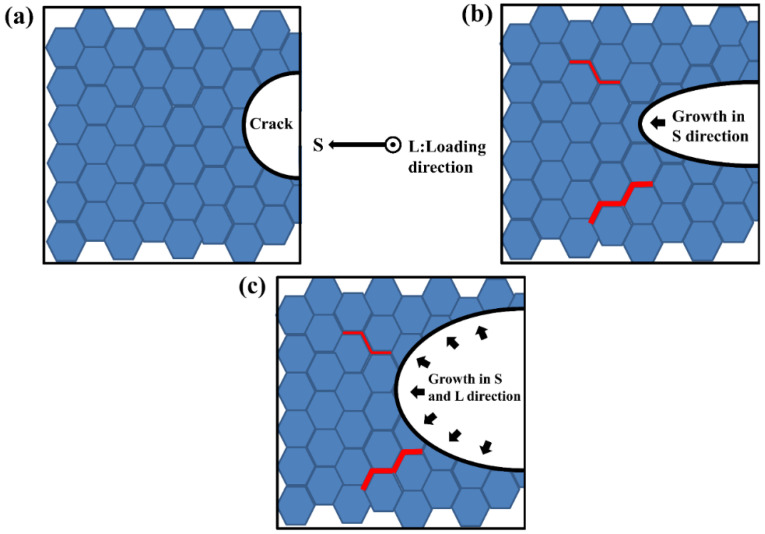
Model 3 of secondary crack evolution in the longitudinal plane during main crack propagation. The red bold line segment is assumed to be secondary microcracks. (**a**) Crack initiation stage; (**b**) Early crack propagation; (**c**) Stage of rapid crack propagation.

**Table 1 materials-15-04698-t001:** Chemical composition of 6005 aluminum alloy and welding wire.

Material	Mass Fraction/%								
	Fe	Cu	Si	Mn	Mg	Cr	Zn	Ti	Al
6005	0.35	0.30	0.60	0.50	0.40	0.30	0.20	0.10	Balance
ER5356	0.40	0.10	0.25	0.35	4.80	0.15	0.10	0.13	Balance

**Table 2 materials-15-04698-t002:** Welding parameters.

Welding Current/A	Welding Voltage/V	Welding Speed/(mm·s^−1^)	Protective Gas/(L·min^−1^)
160	21	5.0	20

**Table 3 materials-15-04698-t003:** Fatigue data table of MIG and BM 6005A-T6 aluminum alloy.

Sample	Ordinal	Stress Range (MPa)	Fatigue Life (Cycle)	State
MIG	1	50	286,854	failure
	2	45	586,235	failure
	3	40	635,117	failure
	4	35	1,364,032	failure
	5	32.5	2,059,690	failure
	6	30	4,076,338	failure
	7	25	10,000,000	runout
	8	20	10,000,000	runout
BM	1	200	302,481	failure
	2	180	531,351	failure
	3	160	795,204	failure
	4	155	600,223	failure
	5	150	2,088,060	failure
	6	145	4,125,728	failure
	7	140	10,000,000	runout
	8	120	10,000,000	runout

## Data Availability

The data that support the findings of this study are available from the corresponding author upon reasonable request.

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
