# Peer review of "Fatigue Property and Small Crack Propagation Mechanism of MIG Welding Joint of 6005A-T6 Aluminum Alloy"

_materials, 2022, doi:10.3390/ma15134698_

Round 1

Reviewer 1 Report

Authors insist that this paper proposes the results for fatigue fracture and small crack mechanism of 11 6005A-T6 aluminum alloy MIG welded joints. Especially, they mentioned as follows:

1) the impacts of the chief crack propagation on the subordinate cracks and the small-scale mechanism of the weld.

2) effects for stress concentration that short hardness and lap joints and uncomplicated to fracture at the weld.

3) for the growing phase of fatigue cracks, impurities obstructed the crack growing and alterd the way of crack growing. The growing of tiny cracks was prevented and obstructed by big cracks, so that the growing of tiny cracks was ceased.

I recommend to accept after minor revision. My judgments come from as following:

1)    This result of this papers is a good example to explain well why crack is generated and propagated. However, abstract, introduction, and conclusion are short and not concise. Authors need to be modified.

2)    Authors also combine section 3 & 4 into section 3 with title “results and discussion”.

Author Response

感谢您对我的文章提出宝贵意见,请看附件。

Reviewer 2 Report

The submitted manuscript discusses the fatigue property and crack propagation mechanism of MIG welded 6005A-T6 Al alloy. The manuscript topic is related to the journal scope. However, it seems not prepared well for submission and contains many inconsistencies. The reviewer's report was made to improve the quality of the submisison as per attached. The pdf file contains all comments and authors are recommended to answer them all. General comments are:

1- Abstract does not reflect the entire work preciesly, it only presents some results. Authers are recommended to rewrite the abstracct.

2- The references are not typed properly in the text. The author's first name should not appear in the text. The reference style must be revised in the entire paper.

3- The literature review is poorly written and the given information is insufficient to find the scientific contribution of this article. Authors are strongly requested to revise the literature data. A recently published work on the fatigue behavior of AA6005 alloy was found in the literature but not included in this paper.

4- The resolution of the figures is very poor and must be improved to explain the results properly.

5- The discussion section is very short and does not discuss all results in the paper.

6- The conclusions do not support the results and must be enhanced.

7- The authors must use a scientific language when describing the results. It was noticed that the writing language and style is more general and does not add to the science that much.

Specific comments can be found in the attached file.

Author Response

hank you for your valuable comments on my article,please see the attachment.

Reviewer 3 Report

The manuscript "Fatigue Property and Small Crack Propagation Mechanism of MIG Welding Joint of 6005A-T6 Aluminum Alloy" has been reviewed. It's an experimental research on the fatigue fracture and small crack mechanism of 6005 T6 Al MIG welded joints.

The manuscript is interesting, despite not new in absolute. English is almost clear however it should be revised by english native speaker.

It can be reconsidered for the acceptance at least after the following major revisions.

The novelty aspects of the research, in relation to the state of the art, should be better highlighted in abstract, introduction and conclusions.

Line 38. Double full stop!

Line 74: etched, not corroded.

Line 89: please motivate the reason for R= 0.1 and the (high frequency) 20 HZ.

Fig. 1, 3, 4 and 5. Increase text size and overall readability!

Fig. 5. Please specify why double log curve has been used. Usually the wohler curves do not show the stress/load in a log scale. Please change or explain!

Why the authors do not consider to perform PWHT on the welded joints?

The lower mechanical properties of the joints are also due to the absence of precipitates in the WZ and this aspect should be discussed and taken into account.

Finally in the conclusions a possible solution for increasing mechanical properties should be introduced.

References [18] and [19] are not called on the main text.

References (all) are not in compliance with the journal requirements.

Author Response

(The authors gave the same response as above.)

Round 2

Reviewer 2 Report

The authors respponded to all given comments and the manuscript can be accepted after minor corrections. The minor corrections include some writing formats, such as that in Line 48 (double dot at the end of the sentence), Line 54 (different font), Line 59 (no captital after the comma), Line 152 (space after Sa), Line 152 (double space before k must be modified), Line 185 (Delta K must be Italic), the inset of Figure 9 (a) is still not clear, etc. The discussion part must be improved.

Reviewer 3 Report

The manuscript has been significantly improved and can be accepted in the present form.

Author Response

Errors in the article language have been modified.Thank you very much for your valuable comments on the article.